# Failure of Autophagy in Pompe Disease

**DOI:** 10.3390/biom14050573

**Published:** 2024-05-13

**Authors:** Hung Do, Naresh K. Meena, Nina Raben

**Affiliations:** M6P Therapeutics, 20 S. Sarah Street, St. Louis, MO 63108, USA; hung.do@m6ptherapeutics.com (H.D.); naresh.meena@m6ptherapeutics.com (N.K.M.)

**Keywords:** Pompe disease, autophagy, lysosome, muscle, glycogen degradation, enzyme replacement therapy

## Abstract

Autophagy is an evolutionarily conserved lysosome-dependent degradation of cytoplasmic constituents. The system operates as a critical cellular pro-survival mechanism in response to nutrient deprivation and a variety of stress conditions. On top of that, autophagy is involved in maintaining cellular homeostasis through selective elimination of worn-out or damaged proteins and organelles. The autophagic pathway is largely responsible for the delivery of cytosolic glycogen to the lysosome where it is degraded to glucose via acid α-glucosidase. Although the physiological role of lysosomal glycogenolysis is not fully understood, its significance is highlighted by the manifestations of Pompe disease, which is caused by a deficiency of this lysosomal enzyme. Pompe disease is a severe lysosomal glycogen storage disorder that affects skeletal and cardiac muscles most. In this review, we discuss the basics of autophagy and describe its involvement in the pathogenesis of muscle damage in Pompe disease. Finally, we outline how autophagic pathology in the diseased muscles can be used as a tool to fast track the efficacy of therapeutic interventions.

## 1. Background: The Lysosome and Autophagy

The term “autophagy” was introduced in the 1960s by Christian de Duve who had discovered the lysosome and proposed an entirely new concept of the lysosome as a degradative organelle [1]. The term (literally meaning “self-eating”) was adopted to highlight the difference between the lysosomal delivery of intracellular components from the materials that are taken up from extracellular space—which de Duve called “heterophagy” (endocytic pathway). The field of autophagy research remained dormant for three decades and relied largely on morphological studies until the discovery of autophagy-related genes and the core molecular machinery involved in the process [2]. Since the 1990s, the field has literally exploded and became one of the most studied areas in biomedical science with an ever-expanding number of publications.

Three morphologically distinct types of autophagy, all of which involve the delivery of cytoplasmic materials into the lysosomal lumen for degradation and recycling, are recognized in mammalian cells—macroautophagy, chaperone-mediated autophagy (CMA), and microautophagy. The most studied type, macroautophagy (commonly referred to as autophagy), is unique among the three types in that it involves the de novo formation of a transient double membrane vesicle, the autophagosome. The process starts with the appearance of a cup-shaped membrane structure, called the phagophore [3], which surrounds a portion of cytoplasm, expands, and closes to form a double-membrane autophagosome. Autophagosomes fuse with early/late endosome-forming amphisomes [4] or with lysosomes forming autolysosomes, where the inner membrane and the sequestered cargos are broken down by lysosomal hydrolases. The products of lysosomal degradation—amino acids, monosaccharides, free fatty acids, and other building blocks—are transported to the cytoplasm and reused in a variety of biosynthetic processes, as reviewed in [5,6]. The autophagic process culminates with the reformation of fully mature functional lysosomes [7].

In CMA, a subset of soluble cytosolic proteins carrying the KFERQ-like motif bind to a chaperone (the heat shock protein Hsc70) which delivers the substrate protein to the lysosomal surface where the protein–chaperone complex interacts with its receptor, the lysosome-associated membrane protein type 2A (LAMP-2A); this step is followed by protein unfolding and translocation of the substrate into the lysosome, a process mediated by a luminal form of Hsc70 [8].

Microautophagy—a process which was originally defined as the direct incorporation of cytoplasmic contents into the lysosomal lumen—received much less attention. However, recent studies re-defined the process to include lysosomal/endosomal membrane protrusion and invagination, thus giving rise to the emerging conceptual framework of microautophagy as “a unified form of membrane dynamics” [9,10].

In this review, we will focus on macroautophagy, since this type of autophagic process is impaired in Pompe disease, as well as in many other lysosomal storage disorders [11,12,13]. Macroautophagy (thereafter referred to as autophagy) was originally defined as a cellular survival mechanism under starvation; nutrient-starved cells respond by utilizing their own resources to provide amino acids for protein synthesis and energy production. Starvation triggers an acute autophagic response, and it is considered a nonselective (bulk) degradation process by which random cytosolic components are engulfed into autophagosomes. During autophagy, a cytosolic pool of ubiquitous microtubule-associated protein light chain 3 (LC3-I) is conjugated to phosphatidylethanolamine to form LC3-II, which is recruited to autophagosomal membranes and remains there all the way until autophagosomes fuse with lysosomes forming autolysosomes. The two LC3 forms can be easily detected with immunoblotting, thus providing a reliable method for monitoring lysosomal turnover of LC3-II, the marker of autophagosomes [14,15].

Apart from the cellular response to starvation, autophagy is also recognized as a highly selective cellular clearance pathway, by which damaged or worn-out organelles, aberrant protein aggregates, and pathogens are removed, thus contributing to cellular quality control [16,17]. Unlike nonselective autophagy, selective autophagy relies on various receptor proteins that recognize the specific cargo and engage autophagic machinery to direct the cargo for degradation.

The mammalian polyubiquitin-binding protein p62/SQSTM1, a classical selective autophagy receptor, mediates the degradation of ubiquitinated protein aggregates through direct interaction with LC3 located on the isolation membrane/phagophore [18,19]. In addition to p62/SQSTM1, multiple other autophagy receptors, also called adaptors, have been identified [20]. Based on the type of cargo, such as, for example, aggregated proteins, invading bacteria, or damaged organelles, selective autophagy has been classified as aggrephagy, xenophagy, mitophagy, ER-phagy, ribophagy, etc.

Of note, damaged lysosomes can also be eliminated via the autophagic pathway. Years ago, while analyzing LC3/LAMP1 (Lysosome-associated membrane protein 1)-immunostained muscle fibers from patients with Pompe disease, we often observed misshapen lysosomes within the autophagosomes and suggested the term “lysophagy” to describe autophagic elimination of damaged lysosomes [21]. The molecular mechanism of this type of selective autophagy has gained great interest, and a recent study demonstrated that in both HeLa cells and neurons, p62/SQSTM1 is recruited to damaged lysosomes and serves as a lysophagy adaptor in a process regulated by the small heat shock protein HSP27 [22].

Over a decade ago, the landmark discovery of transcription factor EB (TFEB) represented a major turning point in the lysosomal field: multiple genes encoding lysosomal enzymes and lysosomal membrane proteins were shown to be coordinately transcribed and regulated by TFEB which binds to a palindromic 10 bp motif (called the coordinated lysosomal expression and regulation motif; CLEAR) in their promoters [23]. Later on, a closely related transcription factor E3 (TFE3), that belongs to the same MiT-TFE family of basic helix–loop–helix leucine-zipper transcription factors, was shown to have a similar effect [24].

Furthermore, these two transcription factors drive expression of genes involved in both lysosomal and autophagosomal biogenesis and promote autophagosome–lysosome fusion and lysosomal exocytosis [24,25,26]. The importance of these findings can hardly be overestimated; most notably, lysosomes are not just the recipients of autophagic cargos but, rather, are active participants in the autophagic process. Coordinated regulation of lysosomal–autophagosomal biogenesis makes perfect sense—during starvation, the expansion of autophagosomes should be matched with an expansion of lysosomes to enable complete degradation of autophagic cargo.

TFEB and TEF3 shuttle between the cytosolic surface of lysosomes and the nucleus depending on their phosphorylation state. When phosphorylated, TFEB and TFE3 remain inactive in the cytosol, but under starvation or other stress conditions, they are dephosphorylated and translocated to the nucleus, thereby inducing the expression of multiple autophagy- and lysosome-related genes (for review, [27]). The mechanistic target of rapamycin complex 1 (mTORC1), a master regulator of cell growth, is a major kinase responsible for TFEB and TFE3 phosphorylation [28,29]. Inhibition of mTORC1 under nutrient-poor conditions, coupled with lysosomal Ca^2+^ release through mucolipin 1 (MCOLN1), activates a calcium-dependent phosphatase, calcineurin, which dephosphorylates TFEB/TFE3 and promotes their nuclear translocation and the induction of lysosomal and autophagic genes [30] (reviewed in [31,32]). 

In mammalian cells, the initiation of autophagy depends on the activation of ULK1 (unc-51 like autophagy activating kinase 1)—one of the most upstream proteins of the core autophagy machinery [33]. The activity of the ULK1 complex is tightly regulated by the opposing effect of mTORC1 and a key energy sensor, AMP-activated protein kinase (AMPK). AMPK directly stimulates autophagy via phosphorylation of ULK1 (at Ser 317 and Ser 777), whereas mTORC1 inhibits autophagy by phosphorylating ULK1 at different sites, thus preventing the AMPK–ULK1 interaction [34,35]. Furthermore, AMPK promotes autophagy indirectly by inhibiting mTORC1 at the lysosomal surface via the TSC2/Rheb axis (tuberous sclerosis complex/Ras homolog enriched in brain); AMPK-mediated activation of TSC2^S1387^ inhibits Rheb, a resident on the cytoplasmic surface of the lysosome and the master activator of mTORC1 [35,36,37,38]. Starvation inhibits mTORC1, thus initiating autophagy and triggering a transcriptional program required for lysosomal biogenesis (Figure 1). 

Perhaps central to the multiple levels of crosstalk between AMPK and mTORC1 and their opposing effects is that both are activated at the cytosolic surface of the lysosome, which serves as a signaling platform to regulate the balance between catabolic and anabolic processes in the cell [31,39,40].

Thus, our understanding of the role of lysosomes has thoroughly changed: what once was viewed as a simple degradative organelle, lysosomes have emerged as a dynamic control center for cellular metabolism [31,41]. The lysosome is well-positioned to survey the cellular nutrient and energy levels and initiate the signaling pathways, thus allowing the cell to attune to emerging conditions [32,42,43].

## 2. Autophagy in Pompe Disease

### 2.1. Brief Introduction

Pompe disease is caused by a deficiency of the lysosomal acid α-glucosidase (GAA), the sole enzyme responsible for the breakdown of glycogen to glucose in the lysosomal lumen. The deficiency leads to progressive accumulation of glycogen in swollen lysosomes in tissues throughout the body, but the damage to cardiac, respiratory, and skeletal muscle is most evident. Therefore, for years, the disease was viewed as a metabolic myopathy. However, the introduction of enzyme replacement therapy combined with the increased awareness of this rare disorder broadened this description, and nowadays, Pompe disease is recognized as a multisystem neuromuscular disorder with CNS involvement [44,45] (reviewed in [46]).

A complete or near-complete lack of GAA activity is invariably associated with the most severe rapidly progressive infantile-onset form of the disease (IOPD); the most common clinical manifestations include the onset of symptoms in the first months (or even days) of life, feeding difficulties, trouble breathing, enlarged heart, muscle weakness, and head lag. Without treatment, most babies die from cardiac or respiratory complications before their first birthday [47]. All other cases of partial enzyme deficiency and the onset of symptoms of proximal limb-girdle myopathy (usually without the heart involvement) any time after 12 months of age are commonly lumped together into a late-onset form (LOPD); although less severe than IOPD, this form is still a debilitating disorder eventually leading to wheelchair dependency and respiratory failure [44,48]. This broad definition of the late-onset subtype is somewhat problematic because the appearance of symptoms in a young child can hardly be called “late-onset”. Perhaps a better definition of clinical phenotypes, adopted by several groups, is ‘classic infantile’, ‘childhood’, and ‘adult’ Pompe disease [49,50,51]. The commonly cited frequency of the disease is 1 in 40,000 live births, but the results of newborn screening make this estimate outdated and reveal a much higher frequency [52,53,54].

Acid α-glucosidase, like dozens of other lysosomal hydrolases, is a glycoprotein that traffics to the lysosomal system through the mannose-6-phosphate (M6P) receptor-mediated pathway [55,56,57,58]. The newly synthesized 110-kD GAA precursor protein is glycosylated (the addition of sugar side chains to certain asparagine residues) in the endoplasmic reticulum (ER) and delivered to the Golgi complex where it is modified by the addition of M6P moieties; the M6P residues serve as recognition signals for mannose-6-phosphate receptors (MPRs) that transport the protein precursor to early endosomes in clathrin-coated vesicles which bud from the trans Golgi network; at the low pH of the late endosome, the M6P-tagged protein and the receptor part ways—the receptors are recycled back to the Golgi, whereas late endosomes fuse with lysosomes, releasing the enzyme to complete its maturation at the acidic pH of the lysosome. Within late endosomes/lysosomes, the enzyme undergoes extensive proteolytic and carbohydrate processing producing fully processed 76- and 70 kD mature lysosomal forms with high affinity for glycogen [59,60]. Importantly, a portion of the precursor protein can be secreted, taken up by neighboring cells via MPR receptors on the cell surface, and delivered to the lysosome through the endocytic pathway—a process that provides the basis for enzyme replacement therapy (ERT) for Pompe disease and other lysosomal storage disorders.

This type of therapy—ERT with recombinant human GAA (rhGAA; alglucosidase alfa; Myozyme^®^, Sanofi, Paris, France; offered as Lumizyme in the US) was approved by the European Medicines Agency and by the US Food and Drug Administration in 2006 for long-term treatment of Pompe disease patients across the clinical spectrum.

### 2.2. Role of Autophagy in Skeletal Muscle Damage in Pompe Disease

The early research on autophagy in Pompe disease grew out of preclinical studies testing the efficacy of Myozyme in a knockout mouse model (KO) [61]. The treatment successfully reversed cardiac pathology and reduced cardiac glycogen to normal levels. In contrast, the uptake of the enzyme in skeletal muscle was much lower compared to that in the liver and heart, muscle glycogen reduction was modest at best, and some fibers, particularly fast-twitch glycolytic type II myofibers, showed little or no glycogen clearance [62]. Paradoxically, in untreated KO, the heart accumulates significantly more glycogen compared to skeletal muscle, indicating that the therapeutic effect does not exclusively depend on the amount of storage material.

Electron microscopy of predominantly type II muscle of KO mice revealed large areas of autophagic debris—double-membrane autophagic vacuoles with undigested materials, multivesicular bodies, concentric multimembrane electron dense structures, etc.—in addition to typical enlarged glycogen-filled lysosomes. Similar structures, called “non-contractile inclusions”, were observed in muscle from another mouse model of Pompe disease [63]; the mechanical effect of these inclusions appeared to contribute to a decline in muscle performance [64]. Furthermore, the morphological observation of large pools of autophagic debris in muscle biopsies from adult Pompe disease patients had been reported way back in 1970 by Andrew G. Engel [65], but at that time, the role of autophagy and the autophagy machinery—the proteins and the sequence of events required for the completion of the autophagy pathway—were not known.

By the time of preclinical testing and clinical trials with Myozyme, the field of autophagy had come a long way, and an array of tools to monitor different steps of the process became available. Apart from academic interest in the subject, autophagic pathology in the diseased muscle appeared to have serious practical consequences: autophagic accumulation in skeletal muscle (but not in cardiac muscle) of KO mice was linked to the resistance to therapy [62,66,67,68].

The results of ERT clinical trials as well as follow-up and investigator-led long-term studies aligned with preclinical data. Treatment with alglucosidase alfa resulted in rapid and impressive reversal of cardiac abnormalities in infants; infantile form of the disease once guaranteed an early death, but the therapy has given many patients a chance to live much longer [69,70,71]. However, optimism was clouded by reality: the therapy did not fully halt or reverse skeletal muscle pathology in long-term survivors with the infantile form of the disease, and most adults experienced a steady decline after the initial improvements over the first couple of years on ERT [48,72,73,74].

So, the researchers went back to the basics and analyzed muscle cells derived from KO mice as well as muscle biopsies of Pompe disease patients. Lysosomal pH measurements and transfection of KO myoblasts with markers for lysosomes (LAMP1), early endosomes (Rab5 and EEA1), late endosomes (LAMP1/CI-MPR), and autophagosomes (LC3) revealed defective acidification of a subset of enlarged lysosomes, expansion of all vesicles of the endocytic and autophagic pathways, and their decreased mobility. Confocal microscopy of LAMP1/LC3 immunostained single myofibers isolated from muscle of KO mice—an approach best-suited for the detection of autophagic buildup—revealed the full extent of pathology in therapy-resistant fibers. The buildup area often spans the entire length of the fiber (with or without interruption) and exhibits a distinct pattern of myofibrillar disarray and a profoundly disorganized microtubule network [67].

Autophagy defects are a common feature of many lysosomal storage diseases [11,12,13], but the sheer volume occupied by the autophagic buildup in KO muscle is truly remarkable (up to 40% in some fibers). Muscle pathology from KO mice appeared something akin to that in a group of autophagic vacuolar myopathies despite large differences in their clinical presentations and etiology [75,76]. Autophagic buildup, never seen in normal muscle, can be detected in diseased muscles as early as in 10-day-old animals, and it greatly expands with age [77]. Furthermore, confocal microscopy of live-cultured single muscle fibers exposed to fluorescently labeled Myozyme showed that a significant portion of the therapeutic enzyme was diverted away from lysosomes and trapped in the buildup areas without resolving it [78,79,80]. This entrapment is not surprising since the exogenous recombinant enzyme destined for the lysosome traffics along the endocytic pathway which converges with the autophagic pathway at the stage of early and late endosomes [4].

As in mouse models, autophagic buildup, filled with potentially toxic ubiquitinated protein aggregates, glycogen particles, autophagy substrate p62/SQSTM1, and lipofuscin (an indicator of oxidative damage and mitochondrial dysfunction), is a prominent feature in muscle from patients with the disease; in some fibers, particularly from biopsies of adult patients, the enlarged lysosomes in the surrounding buildup-free areas look like innocent bystanders compared to the autophagic pathology [21,68,81,82]. Taken together, these data unequivocally establish dysfunctional autophagy as a major secondary abnormality in the diseased muscle that leads to muscle damage and impairment of muscle function.

### 2.3. The Underlying Mechanisms of Defective Autophagy

The autophagic pathway is a multistep process that involves initiation, autophagosome formation, fusion with the lysosome, cargo degradation, and the lysosomal efflux of raw materials. The whole process is defined as autophagic flux—a measure of autophagic degradation activity [83]. Initiation of autophagy is mediated by the ULK1 and the autophagy-specific class III phosphatidylinositol 3-kinase complex I (PI3K), which contains Beclin1, ATG14, vacuolar protein sorting 15 (Vps15), and Vps34. We and others reported an increase in AMPK-mediated phosphorylation of ULK-1^S317^ and elevated levels of Vps15/Vps34/Beclin1 in muscle biopsies from KO mice and Pompe patients, indicating activation of autophagy [77,84].

The involvement of the mTORC1 (inhibitor of autophagy) and AMPK (activator of autophagy) pathways in the pathophysiology of muscle damage was analyzed in GAA-deficient multinucleated myotubes (which replicate the enlargement of glycogen-filled lysosomes and defective autophagy [80]) and in whole muscle from KO mice. Phosphorylation of the two major downstream mTORC1 targets, the 4E-BP1 (eukaryotic translation initiation factor 4E-binding protein 1) repressor protein and ribosomal protein S6 kinase 1 (S6K1), was decreased (the ratios of p-4E-BP1^S65^/total and p-S6K^T421/S424^/total) suggesting a reduction in mTORC1 activity. S6K-mediated phosphorylation of ribosomal protein S6 (p-S6^S235/236^/total) was also decreased, supporting reduced mTORC1 activity. In contrast, the levels of phosphorylated AMPK (p-AMPK^T172^) and its downstream targets, TSC2^S1387^ and p-ACC^S79^, were markedly increased in KO muscle cells. Furthermore, an elevated ADP/ATP ratio was documented in KO muscle, indicating an energy deficit, known to activate AMPK [77,85,86]. Thus, AMPK activation promotes autophagy along two tracks by activating ULK1 and inhibiting mTORC1.

However, a major mechanism underlying autophagic buildup is an impairment of autophagosomal–lysosomal fusion, a condition known as autophagic block. Time-lapse confocal microscopy of KO live-cultured muscle fibers in which lysosomes and autophagosomes were labeled with mCherry-LAMP1 and GFP-LC3 revealed critical insights into the composition and vesicular movement within the buildup areas: a striking paucity of lysosomes suggested an impairment of lysosomal biogenesis, and lysosomal–autophagosomal fusion events were essentially nonexistent. Thus, a rather unfortunate combination of induction of autophagy and autophagic block accounts for massive buildup in the diseased muscle. Overexpression of TFEB in mCherry-LAMP1/GFP-LC3-labeled fibers resulted in near complete elimination of autophagic buildup; importantly, this was the first indication that the buildup *can* be reversed [80]. Yet another study established the feasibility of reversing the fully formed autophagic buildup. Activation of mTORC1 by genetic inhibition of TSC2 boosted protein synthesis, reversed muscle atrophy, and effectively eliminated autophagic buildup in KO muscle [85].

The information on the TFEB activity in Pompe muscle is limited to a single study, in which muscle biopsies of three untreated patients were analyzed. Two consecutive biopsies, with an interval of 6–9 years, from each of the three patients were available for the study. The increase in the phosphorylated form of TFEB (inactivation) was observed in the second biopsy from each patient, suggesting an association with disease progression. Furthermore, there seemed to be a correlation between the degree of TFEB inhibition and the severity of muscle damage [84].

### 2.4. Glycogen Traffic to the Lysosome

Autophagy is a presumptive pathway for glycogen transport to the lysosome. The notion is based on the very definition of autophagy—lysosomal degradation of intracellular materials. Pompe disease underscores the importance of this route of glycogen degradation, particularly for preserving skeletal and cardiac muscle function. However, the questions arise as to which form of autophagy, and if there is a crosstalk between lysosomal glycogen degradation and the well-established canonical cytosolic glycogenolysis under physiological conditions? Early reports provided morphological evidence of the involvement of autophagy in glycogen delivery to the lysosome. Enlarged glycogen-filled autophagic vacuoles were observed in skeletal muscle of neonatal rats; the term “autophagic vacuoles” points to the involvement of macroautophagy. These vacuoles were not seen in fetal tissues and their number rapidly declined within the first few days after birth, suggesting a surge in lysosomal glycogen degradation through macroautophagy to meet the demand for energy in the postnatal period [87]. Similar structures were also observed in the liver and heart of rats in the early postnatal period (when the trans-placental nutrient supply is suddenly interrupted), again supporting the essential physiological role of lysosomal glycogen degradation at a time of high glucose demand [88,89,90,91].

To address the role of macroautophagy in the delivery of glycogen to lysosomes in skeletal muscle, a muscle-specific autophagy-deficient mouse model of Pompe disease was generated via inactivation of a critical autophagic gene, Atg7, in a tissue-specific manner (Atg7GAA double knockout; DKO) [92]. This approach is a conventional way of assessing the role of autophagy in a particular tissue since inactivation of autophagy in the whole body is lethal [93,94,95]. Genetic suppression of macroautophagy (once again, thereafter referred to as autophagy) in muscle of DKO mice resulted in an impressive reduction, but not elimination, of lysosomal glycogen accumulation, suggesting that autophagy may not be the only route of glycogen transport to the lysosome; microautophagy appears to be the most likely candidate. Nevertheless, autophagy undoubtedly plays a prominent role in glycogen delivery to lysosomes. Of note, the absence of autophagic buildup in skeletal muscle of KO mice combined with reduced glycogen burden rendered this tissue amenable to Myozyme treatment [92].

Glycogen particles are commonly seen inside enlarged double-membrane autophagosomes in diseased muscles. Given the cytoplasmic scattering of glycogen particles, their presence in autophagosomes could be a result of a nonselective process. Preferential entrapment of poorly branched glycogen into autophagosomes was suggested [96,97], but these studies still need further clarification. However, the concept of selective glycogen autophagy, called glycophagy [98], recently gained a lot of interest. In the context of Pompe disease, an interesting and thought-provoking terminology was recently used by Heden and colleagues to define the illness as a “long-term glycophagy deficiency” (as opposed to a short-term glycophagy deficiency under experimental conditions in vitro) [99].

The starch-binding domain-containing protein 1 (STBD1) was shown to contain both the carbohydrate-binding domain and a potential interacting motif that is required for binding to the cognate autophagy adaptor protein GABARAPL1 (gamma-aminobutyric acid A receptor-associated protein), a member of the LC3/GABARAP protein family of autophagosome-localized proteins (mammalian orthologs of Atg8 in yeast) [98,100]. Therefore, STBD1 was proposed as a novel receptor (cargo binding protein) for anchoring glycogen to the autophagosomal membrane. The implications of this discovery for Pompe disease research were obvious: if STBD1 is, indeed, responsible for glycogen transport to lysosomes, its inhibition would reduce glycogen burden in the diseased muscle, thus making this protein a new therapeutic target. However, adeno-associated virus (AAV)-mediated knockdown of STBD1 in KO mice did not reduce lysosomal glycogen accumulation in skeletal muscle, suggesting STBD1-independent glycogen delivery to lysosomes in this tissue [101]. These data were supported by the lack of glycogen reduction in cardiac and skeletal muscles of GAA/STBD1 double knockouts, but, unexpectedly, glycogen accumulation was significantly reduced in the liver, suggesting a tissue-specific role of Stbd1 in glycogen transport [102].

Several other glycogen-related proteins that also contain the LC3/GABARAP-interacting motif to incorporate the cargo into autophagosomes were identified in a proteomic analysis of the autophagy interaction network and in silico, such as glycogen-branching enzyme, glycogen phosphorylase, and the muscle form of glycogen synthase, but the presence of this motif does not guarantee the actual binding and anchoring of glycogen molecules to the autophagosomal membranes [103,104,105]. Interestingly, glycogen synthase was shown to regulate glycogen autophagy in skeletal muscle through its interaction with Atg8 protein in a chloroquine-induced Drosophila model of vacuolar myopathy [106]. However, a proteomic analysis of the autophagy interaction network in human cells did not validate glycogen synthase as an autophagy receptor [103], and the mechanism of glycogen transport to the lysosome in muscle remains an open question.

On the other hand, it has now become clear that the defective muscle glycophagy in Pompe disease has profound metabolic consequences; there is a plethora of metabolic changes, such as alterations in lipids, amino acids, TCA cycle, and glycogen and glucose metabolism, that have been shown to occur in the diseased muscle in both KO mice and patients [107,108,109,110]. In a recent study, efficient Gaa knockdown in C2C12 myotubes resulted in a significant increase in lysosomal glycogen, thus creating a short-term glycophagy deficiency. In this setting, the cells were shown to activate PPAR, AKT, and AMPK signaling pathways and to rewire their metabolic program to decrease glucose metabolism and increase fatty acid and glutamine metabolism [99]. These data indicate that glycophagy is a highly regulated process involved in maintaining cellular energy homeostasis [99] (reviewed in [105,111]).

### 2.5. Next-Generation Enzyme Replacement Therapy: Effect on Autophagy

The link between defective autophagy and poor skeletal muscle response to alglucosidase alfa (Myozyme) suggested that an additional autophagy-targeting therapy may be needed to achieve efficient reversal of muscle pathology in Pompe disease. Yet another likely possibility was that the chemical structure and properties of the drug itself were not optimal for efficient muscle uptake and lysosomal trafficking. The major problem with alglucosidase alfa is that only a small fraction of the manufactured recombinant enzyme contains the cognate ligand, mannose 6-phosphate (M6P), particularly bis-phosphorylated oligosaccharide (two M6P residues on the same carbohydrate chain) with high affinity for CI-MPR (cation-independent mannose-6-phosphate receptor). Thus, only a tiny portion of the administered enzyme is delivered to the lysosome in skeletal muscle. Further, since the ERT is administered via intravenous infusions, only a small fraction of the drug reaches the interstitial space surrounding skeletal muscle cells which necessitates the presence of bis-phosphorylated oligosaccharides to enable CI-MPR binding at such a low enzyme concentration. The presence of M6P on the carbohydrate chains of lysosomal enzymes is a prerequisite for their efficient CI-MPR-mediated cellular uptake and lysosomal trafficking [112] (reviewed in [113]). In fact, even at the time around the approval of Myozyme, it was an open secret that alglucosidase alfa was poorly phosphorylated with only a small amount of M6P, and the quest for new recombinant human GAA (rhGAA) with higher amounts of M6P was already underway. Two new drugs have been recently approved for Pompe disease treatment, and both were shown to be noninferior to alglucosidase alfa.

Sanofi designed a second-generation replacement enzyme (Nexviazyme^®^; avaglucosidase alfa-ngpt) with the goal of increasing its M6P content. This was achieved via chemical conjugation of synthetic oligosaccharides containing M6P residues onto modified sialic acids on preexisting N-liked carbohydrates of rhGAA (Myozyme) via oxime chemistry. The carbohydrate-remodeled enzyme exhibited a much higher affinity for the CI-MPR. The drug was approved for patients (of 1 year or older) in 2021 following a Phase 3 clinical trial (COMET) [114].

Nexviazyme (oxime-neo-rhGAA) was tested in preclinical studies in KO mice. The treatment (20 mg/kg) resulted in much greater glycogen clearance in skeletal muscle compared to unmodified enzyme and improvement in muscle function/strength in young mice. In older, symptomatic KO (10 month of age), muscle glycogen load was significantly reduced but not eliminated, and only a modest improvement in motor function was achieved even at a higher dose of 40 mg/kg [115].

The COMET study, a Phase 3, randomized, double-blind, multicenter trial comparing safety and efficacy of avalglucosidase alfa (Nexviazyme^®^) vs. alglucosidase alfa in adult patients who never received ERT (naïve; n = 100) showed clinically meaningful improvement in respiratory function, ambulation, and functional endurance with the new drug over alglucosidase alfa and provided evidence of its safety and noninferiority [114,116]. Yet another relatively small noncomparative study (NEO1 and an open-label, multicenter, extension study, NEO-EXT; 24 ambulant adult patients enrolled in NEO1, 19 entered NEO-EXT, and 17 remained on NEO-EXT) reported up to 6.5 years’ experience with avalglucosidase alfa in naïve or switch patients who had already been treated with incremental dosages of the same drug for >/=9 months. The drug was well tolerated and stabilized lung function and the ability to walk, particularly in patients aged <45 years [117,118].

Amicus Therapeutics has developed a next-generation rhGAA (Pombiliti™; cipaglucosidase alfa-atga + Opfolda™ (miglustat)) as a two-component therapy to improve the delivery of the enzyme to skeletal muscle lysosomes [113,119]. Unlike previous attempt to conjugate synthetic M6P-bearing oligosaccharides onto rhGAA, Pombiliti™ is a recombinant human GAA that is naturally expressed with high levels of bis-phosphorylated oligosaccharides. Miglustat is a small molecule (glucose analog) chaperone that is used to prevent denaturation of the rhGAA enzyme while in circulation. The drug was approved in 2023 for adult patients (who are not improving on alglucosidase alfa; switch) following a Phase 3 clinical trial (PROPEL) [120].

The PROPEL study, a Phase 3, randomized, double-blind, multicenter trial compared the safety and efficacy of cipaglucosidase alfa + Opfolda™ (Pombiliti™) vs. alglucosidase alfa in both naïve and ERT-experienced adult patients receiving alglucosidase alfa for ≥2 years. The outcome was impressive in the ERT-experienced group. Those who switched to cipaglucosidase alfa/miglustat (n = 65) showed a significant improvement in the primary efficacy endpoint—change from baseline to week 52 in 6 min walk distance (6MWD), as opposed to no improvement in patients who remained on alglucosidase alfa (n = 30). Furthermore, the patients who switched to the new drug had stable forced vital capacity (FVC)—the key secondary efficacy endpoint—whereas those who continued on alglucosidase alfa showed a reduction in this parameter after an initial small improvement at week 12. In addition, nearly all secondary endpoints numerically favored cipaglucosidase alfa/miglustat over alglucosidase alfa.

However, these results were not replicated in the ERT-naïve cohort: Both treatments (n = 20 cipaglucosidase alfa/miglustat; n = 8 alglucosidase alfa/placebo) showed a similar increase in 6MWD and a similar decline in FVC. When all the data were combined (overall population), cipaglucosidase alfa/miglustat did not meet the prespecified criteria for superiority compared to alglucosidase alfa for improving 6MWD. Although the rationale for combining the data is somewhat questionable (especially considering a very small group of naïve patients on alglucosidase alfa), as of now, Pombiliti™ received marketing authorization for patients who are not improving on alglucosidase alfa. The safety profile was no different between the two drugs. In the ongoing open-label extension of the PROPEL study, any improvements in motor/respiratory function and biomarker levels in patients on cipaglucosidase alfa/miglustat treatment were maintained to week 104 regardless of previous ERT status [121].

The two clinical trials and the extension studies did not require muscle biopsies; therefore, the effect on autophagy was not investigated. However, cipaglucosidase alfa/Miglustat (originally called ATB200/AT2221) was extensively studied and compared to alglucosidase alfa in preclinical trials. ATB200/AT2221 was shown to be unequivocally more effective compared with alglucosidase alfa at reversing both the primary and secondary abnormalities—lysosomal glycogen accumulation and autophagic buildup in muscle tissue. The difference between the two treatments was already apparent after two biweekly administrations of the drugs and became more pronounced after four doses. The effect of ATB200/AT2221 on autophagy was measured by the reduction in the levels of the lipidated form of LC3 (LC3-II) and the autophagy substrate p62/SQSTM1 (a marker of autophagic flux), as well as via confocal microscopy of immunostained isolated single muscle fibers with Lamp1 and LC3. More than 95% of the fibers still contained autophagic buildup despite treatment with alglucosidase alfa, whereas the number of fibers with typical buildup fell to less than 30% in ATB200/AT2221–treated mice [122].

The effect of ATB200/AT2221 was also investigated following long-term treatment for up to five months. The levels of Vps15/Vps34/Beclin 1 (initiation of autophagy), LC3-II, p62/SQSTM1, and Galectin 3 (LGALS3/galectin 3), a marker of lysosomal damage [123], were all back to the WT levels following the therapy. Only occasional fibers (<3–4%) contained typical buildup in ATB200/AT2221–treated mice, a finding that was supported by analysis of unstained muscle biopsies via second harmonic generation (SHG) imaging—the technique that provides structural information on muscle architecture using relatively large bundles of unstained minimally processed muscle tissue [124]. Furthermore, the treatment improved AMPK/mTORC1 signaling as well as metabolic abnormalities associated with the defect in lysosomal glycogen degradation [77].

The study was the first to provide evidence that the autophagic buildup can be reversed with ERT with rhGAA that exhibits efficient lysosomal trafficking and proper endosomal/lysosomal processing, and that an additional autophagy-targeting therapy is not needed. Also, this work experimentally confirmed the prediction that the reversal of lysosomal (primary abnormality) and autophagic (secondary abnormality) defects in the diseased muscle upon treatments would proceed in the same order. Indeed, the comparison of ATB200/AT2221 vs. alglucosidase alfa as well as systemic vs. liver-targeted gene therapy in KO mice demonstrated that complete/near complete glycogen clearance is a precondition for the buildup resolution [122,125]. The mechanism underlying the reversal of the autophagy defect is not exactly clear. However, the increase in galectin-3 in the diseased muscle and its normalization following successful treatments provide a clue. In a recent paper by Jia et al. [126], galectin-3, a β-galactoside-binding cytosolic lectin, was shown to orchestrate the steps of cellular response to endolysosomal damage by ESCRT (endosomal sorting complexes required for transport)-based membrane repair, induction of autophagic removal of damaged lysosomes (lysophagy), and activation of TFEB-mediated lysosomal biogenesis. The generation of a sufficient number of glycogen-free functional lysosomes in the diseased muscle following efficient therapy (ERT or gene therapy) may explain the reversal of autophagic buildup. In addition, glycogen degradation in autophagosomes (with a less acidic environment than in lysosomes) may be more efficient with ATB200/AT2221 compared to alglucosidase alfa. But irrespective of the mechanism, one thing is clear—the elimination of autophagic buildup is a reliable indicator of therapeutic efficacy.

### 2.6. Relieving the Burden of Autophagy as a Gauge of Therapeutic Success

Identification of an autophagosomal marker LC3, a mammalian homologue of yeast Atg8, allowed for the generation of transgenic mice systemically expressing green fluorescent protein fused to LC3 (GFP-LC3 mice) [127]. This mouse line, called “autophagy-monitoring mice” [95], was made to observe and evaluate the basal levels of autophagy and response to starvation by quantitative analysis of GFP-LC3 dots in multiple harvested tissues. These transgenic mice were crossed to Gaa KO mice to generate a GFP-LC3:KO reporter mouse model of Pompe disease [80]. Massive autophagic buildup can be clearly seen in virtually all muscle fibers isolated from limb muscles [extensor digitorum brevis (EDL) and gastrocnemius] of GFP-LC3:KO mice (Figure 2). Therefore, the efficacy of new therapeutic approaches can be evaluated by ex vivo analysis of isolated myofibers or small muscle bundles from these muscle groups.

However, a major advantage of the reporter model is that it enables observation and monitoring autophagy in live anesthetized animals using high-resolution intravital microscopy (IVM). IVM is a powerful optical technique that provides real-time imaging of cellular processes within the natural tissue context at microscopic resolution, such as gene expression and protein activity, cell trafficking and interactions, etc. This approach has been widely used in cell biology, neurobiology, immunology, cancer research, and many other research fields. Unlike typical in vivo studies, which require euthanizing animals and analyzing the tissues of interest ex vivo at different time points, IVM allows repeated longitudinal imaging of the same location in the same animal (reviewed in [128]). In retrospect, intravital microscopy would have been an ideal way to evaluate the dynamic interaction between lysosomes and autophagosomes in muscle of live KO mice rather than time-lapse confocal microscopy of isolated cultured myofibers [80].

To prove that the outcome of a given treatment can indeed be assessed by visualizing the extent of autophagic buildup, gastrocnemius muscle of GFP-LC3:KO was imaged following AAV-mediated systemic gene therapy. The exact same gene therapy in the original KO had been previously shown to reverse both lysosomal and autophagic defects in this tissue—a lengthy process that involved a range of biochemical/molecular biology methods to analyze the samples ex vivo [125].

Autophagic buildup of different sizes, seen in every fiber of untreated GFP-LC3:KO, was not seen in treated animals, and the fibers looked indistinguishable from those in GFP-LC3:WT, thus proving the point (Figure 3, shown for untreated and treated GFP-LC3:KO). Imaging of limb muscle requires surgical exposure of a particular muscle group—a setting that is referred to as an “acute model” (the animal is sacrificed at the end of the session); however, the use of an implantable imaging window/chamber would make it a “chronic model” that permits longitudinal studies [128].

Importantly, in the context of Pompe disease, a naturally chronic setting—noninvasive imaging of the striated tongue muscle was shown to be equally informative. Although less omnipresent, autophagic buildup—a prominent feature in untreated tongue muscle of GFP-LC3:KO—was not seen in gene therapy-treated GFP-LC3:KO animals [129]. Thus, the reporter model of Pompe disease offers a window into the disease progression and enables a highly efficient evaluation of response to therapies by imaging clinically relevant tongue lesion.

## 3. Conclusions

What started as an attempt to understand skeletal muscle resistance to therapy in Pompe disease led to the deciphering complex pathophysiology of muscle damage; autophagic defect is now recognized as a classic characteristic of the disease along with lysosomal glycogen accumulation. With technological advances, including high-resolution in vivo imaging, the autophagy-related pathology can now be exploited for the assessment of new therapies that are under development—new replacement enzymes, various gene therapy strategies, and glycogen substrate reduction therapy.

## Figures and Tables

**Figure 1 biomolecules-14-00573-f001:**
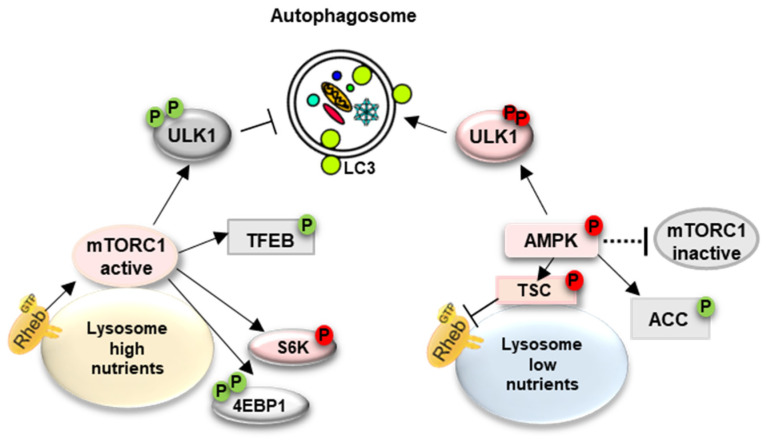
mTORC1 and AMPK signaling. A diagram shows the position of the proteins analyzed in Pompe disease. Under the nutrient-replete condition, mTORC1 is recruited to the lysosome where it is activated by GTP-bound Rheb. Activated mTORC1 suppresses autophagy through phosphorylation-dependent inhibition of the ULK1 complex and the transcription factors TFEB/TFE3; mTORC1 phosphorylates 4E-BP1 (translation repressor protein) and S6K, thereby stimulating the initiation of protein synthesis. Under nutrient deprivation, AMPK promotes autophagy initiation through positive phosphorylation of ULK1. AMPK also promotes autophagy indirectly by inhibiting mTORC1 activity through phosphorylation of TSC2; mTORC1 is displaced from the lysosome (inactivation), and TFEB/TFE3 translocate from cytoplasm to nucleus to promote the transcription of autophagy/lysosome-related genes.

**Figure 2 biomolecules-14-00573-f002:**
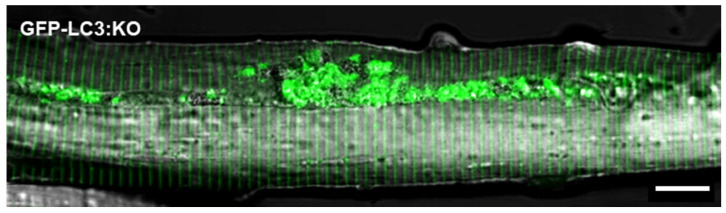
Confocal microscopy of live unstained muscle fibers freshly isolated from a GFP-LC3:KO mouse (ex vivo analysis). The image shows typical autophagic buildup (green area) which disrupts muscle architecture. Bar: 20 μm.

**Figure 3 biomolecules-14-00573-f003:**
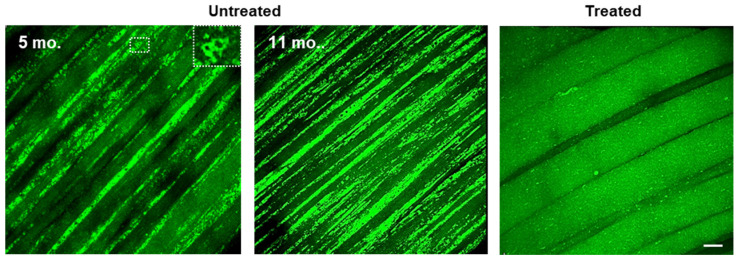
Intravital microscopy of limb muscles of untreated and treated GFP-LC3:KO mice. Gastrocnemius muscles of GFP-LC3:KO mice were imaged at different ages to monitor the progression of autophagic pathology. Inset shows ring-shaped autophagosomes. The right panel shows the elimination of autophagic buildup in a treated GFP-LC3:KO mouse. The IVM imaging was performed two months after a single intravenous administration of an AAV9 vector expressing human *GAA* transgene into a 5-month-old animal. Bars: 30 μm.

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
