# Peer review of "Failure of Autophagy in Pompe Disease"

_biomolecules, 2024, doi:10.3390/biom14050573_

Round 1

Reviewer 1 Report

Comments and Suggestions for Authors

Failure of autophagy in Pompe disease is an excellent review that covers the history and basic mechanisms of autophagy – the pathway responsible for delivering cellular contents to the lysosome for degradation, a brief introduction to Pompe disease, and a summary of the work that has been completed to understand the pathological cascade that occurs in Pompe disease autophagy.

The authors of this manuscript, have significant expertise in this field. In particular the senior/corresponding author, Nina Raben, is the foremost expert on Pompe disease autophagy and is the leading researcher on this topic for nearly 20 years. This review is well written and thorough. It will be an asset to the Pompe research community as a resource on the current understanding of the mechanisms underlying dysfunctional autophagy in Pompe disease. I have no significant critiques of this paper.

Author Response

We thank the reviewer for the evaluation of our manuscript.

Reviewer 2 Report

Comments and Suggestions for Authors

this is a good work on autophagy in pompe disease. I do no have further suggestions, it is ok as it is in my opinion 

Author Response

We thank the reviewer for evaluating our manuscript.

Reviewer 3 Report

Comments and Suggestions for Authors

This review paper is well written in a very professional and exhaustive fashion. I have nothing to add or comment. 

It can be published in the present form. 

Author Response

(The authors gave the same response as above.)
